# How do time trends in inhospital mortality compare? A retrospective study of England and Scotland over 17 years using administrative data

María José Aragón, Martin Chalkley

## ABSTRACT

**Objectives** To examine the trends in inhospital mortality for England and Scotland over a 17-year period to determine whether and if so to what extent the time trends differ after controlling for differences in the patients treated.

**Design** Analysis of retrospective administrative hospital data using descriptive aggregate statistics of trends in inhospital mortality and estimates of a logistic regression model of individual patient-level inhospital mortality accounting for patient characteristics, case-mix, and country-specific and year-specific intercepts.

**Setting** Secondary care across all hospitals in England and Scotland from 1997 to 2013.

**Population** Over 190 million inpatient admissions, either electively or emergency, in England or Scotland from 1997 to 2013.

**Data** Hospital Episode Statistics for England and the Scottish Morbidity Record 01 for Scotland.

**Main outcome measures** Separately for two admission pathways (elective and emergency), we examine aggregate time trends of the proportion of patients who die in hospital and a binary variable indicating whether an individual patient died in hospital or survived, and how that indicator is influenced by the patient's characteristics, the year and the country (England or Scotland) in which they were admitted.

**Results** Inhospital mortality has declined in both countries over the period studied, for both elective and emergency admissions, but has declined more in England than Scotland. The difference in trend reduction is greater for elective admissions. These differences persist after controlling for patient characteristics and case-mix.

**Conclusions** Comparing data at country level suggests questions about the roles performed by or functioning of their healthcare systems. We found substantial differences between Scotland and England in regard to the trend reductions in inhospital mortality. Hospital resources are therefore being deployed increasingly differently over time in these two countries for reasons that have yet to be explained.

Centre for Health Economics, University of York, York, UK

**Correspondence to**
Dr María José Aragón;
mjma504@york.ac.uk

## Strengths and limitations of this study

► The first study to use comprehensive and extensive data on hospital admissions and discharges over a long period of time to study differences in inhospital mortality.

► Establishes a different perspective on inhospital mortality—that of variation across healthcare systems over time—and establishes that two neighbouring countries with otherwise similar healthcare systems have different time paths of inhospital mortality.

► Uses detailed administrative records to control for variation in case-mix and patient characteristics.

► It is not possible to establish the potential causes of the different trends in inhospital reported, but potential causes are established as future avenues of research.

## INTRODUCTION

Inhospital mortality has attracted a good deal of attention and concern when used as a proxy for hospital performance.[1–6] The concern stems from an inability to disentangle consequences of treatment choices from the inherently different risks that patients' medical conditions pose.[7–10] This debate however distracts from the potential knowledge that can be derived by studying inhospital mortality at a more aggregate level.[11] Hospital care is costly and a key resource in addressing a population's healthcare needs. It has been noted that death is a 'core business' of hospitals,[12] and hence understanding how that core business is changing—how much of the 'business' of hospitals it accounts for—is a crucial aspect of health system planning and management.

Without some reference point it is impossible to determine whether an outcome such as declining inhospital mortality is notable or to be expected. Comparing two otherwise similar healthcare systems establishes each as a reference for the other. That reference is more powerful if analysis is conducted in trends. Differences in the levels of inhospital mortality across different jurisdictions could easily be accounted for as the consequence

of unobserved differences between their populations, healthcare needs and service organisation. However, these unobservable factors seem likely to follow common trends, so divergence in the trends of inhospital mortality is more challenging to explain.

This, the first study of its kind, examines the trends in inhospital mortality for England and Scotland over a 17-year period. We establish that 'death as the core business of hospitals' has been declining faster in England than in Scotland over that period.

## METHODS

### Data

In both England and Scotland data are routinely collected on hospital inpatient activity through, respectively, Hospital Episode Statistics and the Scottish Morbidity Record 01. Both data sources report in terms of episodes (period under the care of one consultant), which are then converted into continuous inpatient spells (CIS) corresponding to the period of care that can include transfers within and between hospitals. We construct equivalent measures of CIS for both countries and distinguish between elective (including day cases) and emergency admissions, excluding maternity and regular attenders, using the type of admission of the first episode in the CIS. Both data sources report on the basis of financial years (1 April to 31 March), but for convenience we denote the financial year by its first calendar year. We examine over 190 million CIS from 1997 to 2013 using discharge information to determine whether the patient died in hospital or not.

Both data sources include the characteristics of a patient in regard to age, sex and the deprivations decile of their home address. We use these together with the Healthcare Resource Group (HRG) into which the patient's treatment fell to account for variation in case-mix.

### Empirical methods

The proportion of all CIS that end in death was calculated directly from the data sources, separately for each country, year and admission pathway.

After constructing a binary outcome variable (equal to 1 if the patient died in hospital and 0 otherwise), logistic regression analysis, separately for each admission pathway, was used to determine whether differences across jurisdictions persist after including covariates. To control for the potential influences of patient characteristics, case-mix and socioeconomic circumstances, the covariates included in the analysis were age (using 5-year age bands indicators), sex (as an indicator equal to 1 for female), HRG indicators (there are more than 1000 different HRGs in the data) and deprivation decile indicators (1 being the most deprived and 10 the least deprived). Differences between countries were captured by country-specific dummy variables and interactions between those and year dummy variables.

We ran logit regressions using Stata V.13.

## RESULTS

Figure 1 shows the trend in the inhospital mortality (CIS where patient died/total CIS) for England and Scotland for elective and emergency admissions.

In figure 1 it is apparent that inhospital mortality has decreased in both countries, but has done so more quickly in England than in Scotland in both emergency and elective care. Over the same period the trends of overall mortality, measured by the crude mortality rate (deaths per 1000 population), and life expectancy (in years) have been similar in both countries[13–15] (see figure 2). While overall spending per head on hospital care is higher in Scotland, it has followed a similar (increasing) trend as in England.[16]

Next we describe the relative, England/Scotland, inhospital mortality rate (CIS where patient died/total CIS),

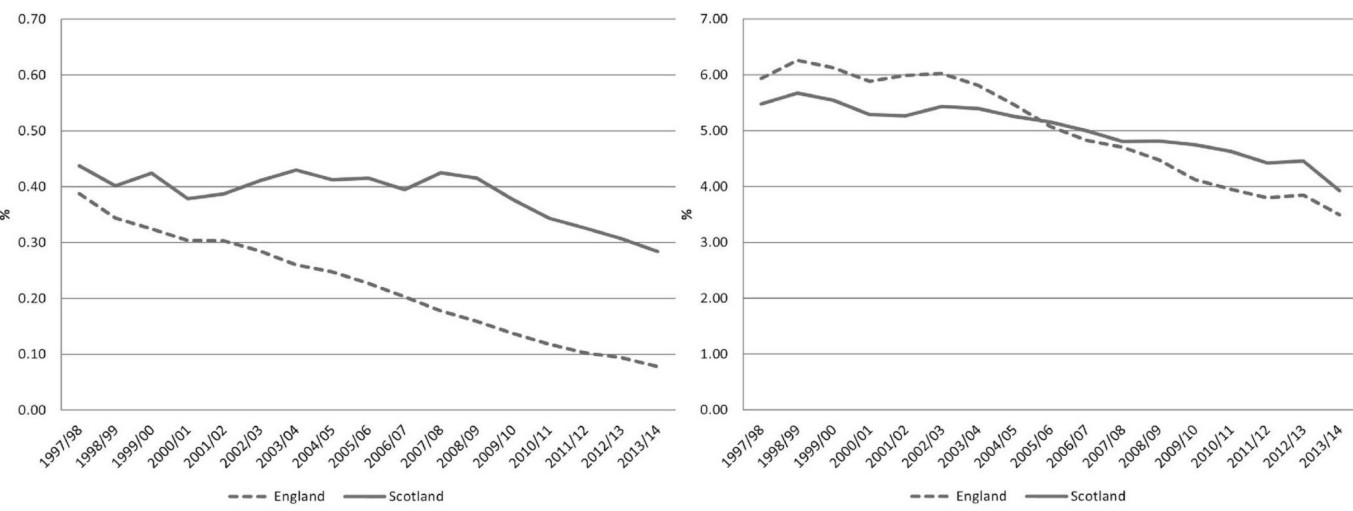

**Figure 1** Inhospital mortality rate. Elective (LHS) and emergency (RHS). The y-axis scales are different; LHS is 1/10 of RHS. LHS, left hand side; RHS, right hand side.

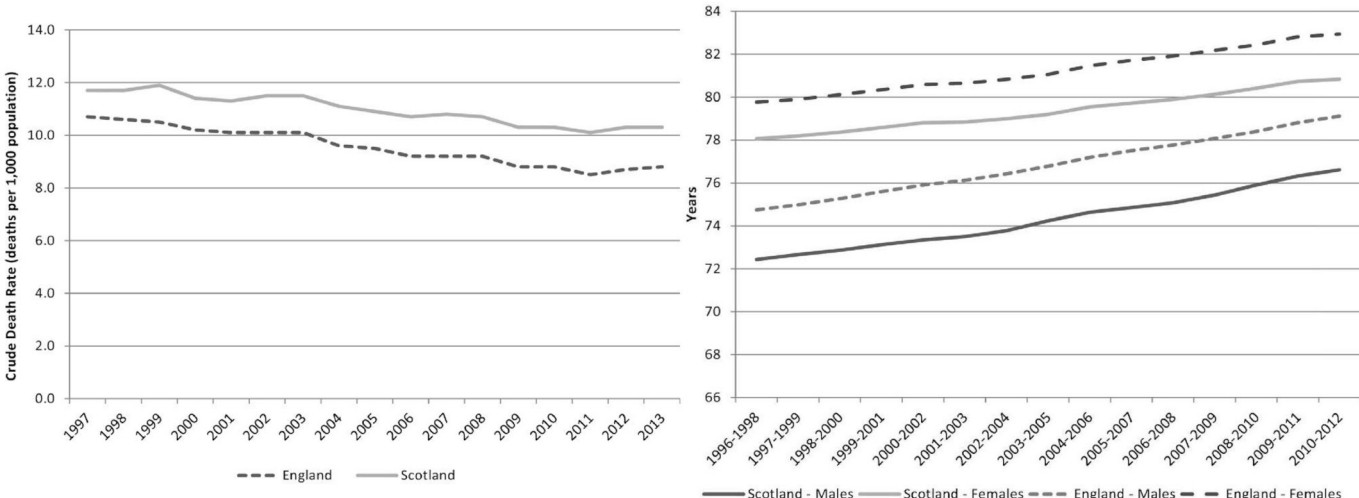

**Figure 2** Crude mortality rate (LHS) and life expectancy (RHS). LHS, left hand side; RHS, right hand side.

again separately for elective and emergency admissions. Figure 3 shows the ratio of the inhospital mortality rates for both countries, normalising to 100 the initial year, and clearly shows the relative change in the inhospital mortality rates.

Next we determine whether these crude, unadjusted differences persist once we account for the different characteristics of the patients who are being treated in the two countries, specifically age, sex, disease proxies and deprivation level.

Table 1 shows the descriptive statistics of the variables used for the regression analysis. Over the period of analysis, inhospital mortality has been higher for emergency than for elective admissions, and emergency admissions' patients are more likely to be men, are younger and more likely to come from the highest deprivation decile than elective admissions.

Table 2 shows the regression results, presented as relative ORs between England and Scotland; this presentation was chosen to simplify the results table and focus on the question of interest: is the reduction in inhospital mortality rates different between the countries after controlling for patient and CIS characteristics?

The results in table 2 confirm what is observed in figures 1 and 3—the reduction in inhospital mortality in England has been faster than that of Scotland throughout the period, even after controlling for patient and CIS characteristics. The results are reported as ORs, showing the relative difference between the two countries in each period; for example, the first row says that in the initial year of the analysis elective admissions were 11% lower in England than in Scotland and emergency admissions were 3% higher. For *electives* England starts with a lower inhospital mortality rate (coefficient in the first row is less than 1 and significant), then there is no clear trend in the difference between the countries until there is no significant difference between the two countries (non-significant coefficient in 2001/2002)

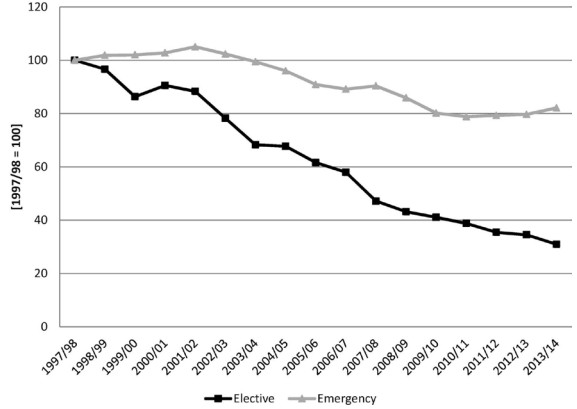

**Figure 3** Inhospital mortality rate. England/Scotland with 1997/1998=100.

| Table 1 | Descriptive statistics | | | |
|---|---|---|---|---|
| | **Mean** | **SD** | **Mean** | **SD** |
| Elective | England | | Scotland | |
| % Died in hospital | 0.21 | 4.62 | 0.40 | 6.32 |
| % Male | 46.86 | 49.90 | 45.31 | 49.78 |
| Age | 54.43 | 21.26 | 53.73 | 21.03 |
| % Decile 1 | 10.26 | 30.35 | 12.06 | 32.57 |
| % Decile 10 | 9.00 | 28.62 | 8.21 | 27.46 |
| Number of observations | 100 945 785 | | 9 886 856 | |
| Emergency | England | | Scotland | |
| % Died in hospital | 4.98 | 21.75 | 5.01 | 21.82 |
| % Male | 48.03 | 50.00 | 49.14 | 49.99 |
| Age | 50.55 | 28.32 | 52.60 | 26.39 |
| % Decile 1 | 14.70 | 35.41 | 15.85 | 36.52 |
| % Decile 10 | 7.11 | 25.69 | 6.21 | 24.13 |
| Number of observations | 74 048 633 | | 8 259 572 | |

| Table 2 | Logit regression results | |
|---|---|---|
| | **Elective** | **Emergency** |
| 1997/1998 | 0.890* | 1.027* |
| | (0.852 to 0.930) | (1.012 to 1.043) |
| 1998/1999 | 0.929* | 1.096* |
| | (0.889 to 0.971) | (1.079 to 1.113) |
| 1999/2000 | 0.898* | 1.105* |
| | (0.859 to 0.938) | (1.088 to 1.122) |
| 2000/2001 | 0.952† | 1.105* |
| | (0.909 to 0.996) | (1.088 to 1.122) |
| 2001/2002 | 0.982 | 1.121* |
| | (0.938 to 1.029) | (1.104 to 1.138) |
| 2002/2003 | 0.941† | 1.095* |
| | (0.899 to 0.986) | (1.078 to 1.111) |
| 2003/2004 | 0.834* | 1.094* |
| | (0.797 to 0.873) | (1.078 to 1.111) |
| 2004/2005 | 0.776* | 1.069* |
| | (0.741 to 0.813) | (1.053 to 1.085) |
| 2005/2006 | 0.727* | 1.029* |
| | (0.695 to 0.761) | (1.014 to 1.045) |
| 2006/2007 | 0.675* | 1.044* |
| | (0.644 to 0.707) | (1.028 to 1.060) |
| 2007/2008 | 0.560* | 1.002 |
| | (0.535 to 0.586) | (0.987 to 1.017) |
| 2008/2009 | 0.534* | 0.933* |
| | (0.511 to 0.559) | (0.919 to 0.947) |
| 2009/2010 | 0.511* | 0.868* |
| | (0.487 to 0.536) | (0.855 to 0.881) |
| 2010/2011 | 0.460* | 0.845* |
| | (0.438 to 0.484) | (0.832 to 0.858) |
| 2011/2012 | 0.403* | 0.854* |
| | (0.383 to 0.424) | (0.841 to 0.867) |
| 2012/2013 | 0.370* | 0.827* |
| | (0.351 to 0.390) | (0.815 to 0.840) |
| 2013/2014 | 0.329* | 0.831* |
| | (0.312 to 0.348) | (0.818 to 0.844) |
| Dummy variables included as controls | | |
| Age group | Yes | Yes |
| Gender | Yes | Yes |
| Deprivation decile | Yes | Yes |
| Healthcare Resource Group | Yes | Yes |
| Number of observations | 110 832 641 | 82 308 205 |

Dependent variable: prob(death). Relative OR England versus Scotland. 95% CIs in parentheses.
* and † indicate 1% and 5% significance, respectively. All regressions include a constant.

and then the difference with Scotland increases over time (coefficient becomes smaller over time) until in the last year inhospital mortality for elective admissions in England is around one-third of that in Scotland. For *emergencies* England started with a higher inhospital mortality rate (coefficient in the top row is greater than 1 and significant), and the difference between the two countries first increased (coefficients become greater) and then decreased until there is no difference between them (non-significant coefficient in 2007/2008) and then England's inhospital mortality rate continues to reduce relative to that of Scotland (coefficients smaller than 1 and significant from 2008/2009 onwards) until being around 27% lower in the last year.

As it can be expected for a study comparing two specific countries, these results are not readily generalisable; the comparison of two specific countries, with similar healthcare systems, will yield a set of results that may or may not correspond to those obtained by comparing any other pair of countries. However, we have established a method of comparison that can apply in any circumstances in which there are suitable data.

## DISCUSSION

This study shows that inhospital mortality for both elective and emergency admissions has been on a 17-year declining trend in both England and Scotland, but that trend reduction has been greater in England. This remains the case after controlling for case-mix and population characteristics.

We have used comprehensive and extensive data on hospital admissions and discharges over a long period of time, providing details of more than 190 million admissions. These data have been adjusted so as to be able to compare two similar healthcare systems so that each can act as a benchmark for the other. While we can establish the differences between the experiences of these two systems with our data, we have not established causal mechanisms for these differences.

Numerous previous studies have examined the variation of inhospital mortality across different hospitals, focusing on the details and limitations of risk adjustment. This study provides a different perspective—that of variation across healthcare systems. While we cannot hope to replicate the detail or depth of previous studies that focus on particular treatments, we do provide a much broader and comprehensive view.

This view suggests a number of important and unanswered questions that have great potential importance for policymakers. Why has the divergence in trend reduction in inhospital mortality developed? In what ways are these two healthcare systems developing different roles for their hospitals? Should there be a concern in Scotland that inhospital mortality is decreasing less slowly and is now substantially higher than in its near neighbour England?

Answering the first of these questions will involve a search for clinical factors that may have exerted a differential impact on inhospital mortality trends in the two countries. There are a number of candidates for such clinical confounders, including, for example, the differential timing of the introduction of screening programme for

high mortality conditions such as abdominal aortic aneurysm,[17 18] and the associated use of endovascular repair. It is worth noting, however, that any one factor is likely to account for only a small fraction of the difference in aggregate trends.

The subsequent questions concern the impact of health system reform and policies once all clinical factors are accounted for. These are also for future research— but we can give some insight and some clues as to the possible answers. One key difference in the development of hospital-based healthcare in Scotland and England over the period studied has been the reform of financing undertaken in England. This has been shown to have resulted in an expansion of activity on a per-capita basis. This suggests that part of the explanation for what we have observed is that hospitals in England are treating more 'less-sick' patients, which would result in a lower propensity for patients to die in hospital simply by increasing the denominator. However, this seems unlikely to be the whole explanation because we have established that the reductions in mortality exist for *both* elective and emergency admissions, and whereas the former would appear susceptible to 'denominator' effect it is less easy to account for emergency admissions in this way. The reduction on inhospital mortality could also be related to the reduction in the duration of hospital admissions (usually called 'length of stay', LoS), which both countries report in the period of analysis.[19 20] We use HRGs to adjust for case-mix; however, HRGs are meant to group together patients with similar diagnosis/treatment and with similar resource intensity.[21] Since we adjusted for the kinds of treatments that are carried out, and the resources needed to deliver them, in England and Scotland it is also difficult to account for the differences in trends in terms of changing case-mix unless our adjustment is substantially flawed because there are large unobserved differences. Basing an analysis on trends mitigates this risk because for it to affect our results requires that the unobserved differences in case-mix between the two systems are changing over time.

This then suggests that there are two avenues to explore further. The first is to determine whether the alternatives to care in hospital setting have diverged in the two countries. If for example alternative settings to which terminally ill patients can be discharged have expanded faster in England than in Scotland, we would observe the kind of differential trend of inhospital mortality established by our analysis. The second, more worrying possibility is that there remains some element of the difference in trend that relates to the efficacy of hospital treatments in the two countries. The details of such potential *quality of care* differences are for clinicians and practitioners who are familiar with hospital treatments of specific conditions to explore, considering any changes in practice or performance targets relevant to them, for example, during the period of analysis Scotland had targets regarding access and treatment of specific patient groups.[22]

**Acknowledgements** The Hospital Episode Statistics are copyright 1997/1998–2013/2014, reused with the permission of the Health & Social Care Information Centre. All rights reserved. The Scottish Morbidity Record data were used with the permission of the Information and Statistics Division Scotland (ISD).

**Contributors** MJA and MC defined the research question and type of analysis required. MJA performed the statistical analysis. MJA and MC analysed the results and wrote the article.

**Funding** This article was developed further to the NIHR-funded project HS&DR-11/1022/19.

**Competing interests** None declared.

**Patient consent** Not required.

**Provenance and peer review** Not commissioned; externally peer reviewed.

**Data sharing statement** No additional data available.

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
