## [Reviewer comments · BMJ Open]

ARTICLE DETAILS

TITLE (PROVISIONAL)	How do time trends in in-hospital mortality compare? A retrospective study of England and Scotland over 17 years using administrative data.
AUTHORS	Aragon Aragon, Maria Jose; Chalkley, Martin

VERSION 1 – REVIEW

REVIEWER	Janet Powell Imperial College London, UK No Competing Interest
REVIEW RETURNED	24-Apr-2017

GENERAL COMMENTS	This review of in-hospital mortality in England and Scotland does not have sufficient depth either in analysis or interpretation. I list below some of the reasons for this: 1 The AAA screening programme for men was rolled out earlier in England than in Scotland2 The in-hospital mortality is much lower for EVAR than for open repair, particularly for elective surgery and as such separate data should be provided for EVAR and open repair, from 2006 when coding changed. Perhaps EVAR has been underused in Scotland? Could the non-intervention rates be higher in England in recent years, to achieve the Quality Improvement Programme targets?3 In-hospital mortality also varies markedly with patient age and sex and this does not appear to have been either analysed or discussed appropriately. Table 1 presumably provides the general population structure and not the AAA population structure? The title of this table does not make this clear, but the age and % males perhaps does.4 In figure 1, the ordinate (vertical axis) for in-hospital mortality rate would appear to have incorrect units for both halves, unless the rate is given per 10,000 population, but this is not specified. Elective in-hospital mortality should be <5% overall and the Vascular Surgical Society Quality Improvement programme expects all units to have a rate of <3.5% for 30-day mortality.5 Do patients stay in hospital longer in Scotland and therefore die in-hospital rather than at home within 30-days?6 Again, the units of the vertical axes need to be specified in Figure 2 and much better described in Figure 3.
---

	7 Finally, separating elective from emergency admissions in HES can be quite difficult and needs to use several different pieces of information, including the method of admission and date of surgery. It is not clear that this has been followed.
--	--

REVIEWER	Mathis Heydtmann Glasgow University & Royal Alexandra Hospital Paisley Corsebar Road Paisley PA2 9PN I live in Scotland and have done epidemiological studies on trends in alcoholic liver disease hospitalisations. Otherwise I don't have any competing interests.
REVIEW RETURNED	01-May-2017

GENERAL COMMENTS	This article addresses hospitalisation mortality which is of great health economic importance in the current climate of austerity. The authors analyse whole population data over 17 years which gives the study significant power to detect long term trends. The methodology including statistics is adequate and overall the article reads well (minor typo page 6/35: "on basis of financial years"), the message is clear, relevant, interesting and provokes further research in that area to explain the differences between the 2 healthcare systems.
---

VERSION 1 – AUTHOR RESPONSE

Reviewer 1

We thank the reviewer for their comments and suggestions. Our responses and the changes made in respect of them are detailed below.

Comment 1 The AAA screening programme for men was rolled out earlier in England than in Scotland
 Comment 2 The in-hospital mortality is much lower for EVAR than for open repair, particularly for elective surgery and as such separate data should be provided for EVAR and open repair, from 2006 when coding changed. Perhaps EVAR has been underused in Scotland? Could the non-intervention rates be higher in England in recent years, to achieve the Quality Improvement Programme targets?

Response: Both of these relate to specific conditions and treatments and how these vary between the two countries. A key point is that our article does not intend to focus on specific conditions, it sets out a comparison of mortality across two entire health care systems over time. We are pleased that the reviewer makes these observations because it is exactly the response that our approach is intended to provoke. There is a wealth of research questions of the kind "to what extent can the difference in treatment of "x" in Scotland or England account for these observed trend differences on in-hospital mortality". Our aim in the article is to provoke such questions, which will naturally come from experts in the respective specific conditions and treatments.

We are the first to suggest the exact nature of the differences that has to be explained, but we do not (and could not) provide the explanations – that is for others. Indeed we explicitly say as much in our discussion:

“Our study cannot answer these questions – that is for future research – but we can give some insight and some clues as to the possible answers.”

Response: We have amended the subsequent section to include possible answers that depend upon the analysis of specific conditions. We think for example that the questions raised under 2. above are a potentially very important subject for further research.

Comment 3 In-hospital mortality also varies markedly with patient age and sex and this does not appear to have been either analysed or discussed appropriately. Table 1 presumably provides the general population structure and not the AAA population structure? The title of this table does not make this clear, but the age and % males perhaps does.

Response: We can confirm that these influences were included in our study. We have clarified, by making an explicit statement in the Empirical Methods sections that the regressions include control variables for age (age group dummies) and sex along with other potential confounders. The coefficients for these variables are not included in Table 2 to simplify presentation (there are 19 5-year-age-group variables). We have amended the text in the table to make it clearer that the control variables are included in the regressions – rather than just state them as dummy variables we label them as controls.

Comment 4 In figure 1, the ordinate (vertical axis) for in-hospital mortality rate would appear to have incorrect units for both halves, unless the rate is given per 10,000 population, but this is not specified. Elective in-hospital mortality should be <5% overall and the Vascular Surgical Society Quality Improvement programme expects all units to have a rate of <3.5% for 30-day mortality.

Response: The plots have as a vertical axis the percentage of spells (CIS) that end because the patient died (CIS ending in death / Total CIS). To clarify this we have added this to the text right before the Figure.

Comment 5 Do patients stay in hospital longer in Scotland and therefore die in-hospital rather than at home within 30-days?

Response: The duration of hospital admissions (length of stay, LoS) has reduced in both countries. Scotland had some specific targets regarding LoS reduction; such targets do not exist in England, but there are adjustments to the Payment by Results (PbR - now National Tariff Payment System, NTPS) tariff based on LoS for some Healthcare Resource Groups (HRGs). We now include some commentary on this in the discussion section.

Comment 6 Again, the units of the vertical axes need to be specified in Figure 2 and much better described in Figure 3.

Response: Thank you. Units added to Figure 2. In Figure 3 there are no units as such, since mortality rates have been standardised to start at 100 but this has been added as explanation to the axis.

Comment 7 Finally, separating elective from emergency admissions in HES can be quite difficult and needs to use several different pieces of information, including the method of admission and date of surgery. It is not clear that this has been followed.

Response: The classification into elective/emergency is based on the classification of the first episode of the CIS. In each data source, every episode has a 'method of admission' (in HES, 1) or 'admission type' (in SMR, 2) which allows the identification of elective and emergency.

1 http://content.digital.nhs.uk/media/23711/Admitted-Patient-Care/pdf/Admitted_Patient_Care_.pdf

2 <http://www.ndc.scot.nhs.uk/Dictionary-A-Z/Definitions/index.asp?Search=A&ID=58&Title=Admission%20Type>

Reviewer 2

Comment: minor typo page 6/35: "on basis of financial years"

Response: Thank you, typo corrected.

VERSION 2 – REVIEW

REVIEWER	Janet Powell Imperial College London UK No Competing Interest
REVIEW RETURNED	22-May-2017

GENERAL COMMENTS	I understand that the authors are viewing the comparison from an health economic perspective across healthcare systems: England and Scotland are rather similar in many respects and do not provoke real contrasts such as a comparison of England and Germany, or the recent case of England versus USA (Karthikesalingam et al NEJM 2016). There also exist comparisons of England versus Sweden for AAA mortality. None of these are either cited or discussed. Therefore, unfortunately I cannot accept that known confounders eg use of endovascular versus open repair and timing of the AAA screening programme roll-out are not even discussed. In particular for elective surgery, in-hospital mortality data from endovascular versus open repair are available from the routine sources used (eg HES). Without such information and discussion against other international comparisons the authors cannot provide the "broad and comprehensive view", which they claim.
---

VERSION 2 – AUTHOR RESPONSE

The reviewer now writes that;

"I understand that the authors are viewing the comparison from an health economic perspective across healthcare systems: England and Scotland are rather similar in many respects and do not provoke real contrasts such as a comparison of England and Germany, or the recent case of England versus USA (Karthikesalingam et al NEJM 2016). There also exist comparisons of England versus Sweden for AAA mortality. None of these are either cited or discussed.

Therefore, unfortunately I cannot accept that known confounders eg use of endovascular versus open repair and timing of the AAA screening programme roll-out are not even discussed. In particular for elective surgery, in-hospital mortality data from endovascular versus open repair are available from the routine sources used (eg HES).

Without such information and discussion against other international comparisons the authors cannot provide the "broad and comprehensive view", which they claim."

Response:

In respect of the first observation that "England and Scotland are rather similar in many respects and do not provoke real contrasts such as a comparison of England and Germany, or the recent case of England versus USA (Karthikesalingam et al NEJM 2016).", we would simply point out that is a key strength of our paper.

Specifically, by comparing two similar systems and finding substantial and enduring differences between them in respect of hospital mortality – differences that have increased over time – we have established a real challenge for practitioners and policy makers to explain what has been observed. If our two systems had been very dissimilar, the challenge might not be such a great one. But England and Scotland share many characteristics including almost exclusive public funding of health care and similar healthcare priorities. Their divergence in regard to hospital mortality needs to be explained both for the assurance of patients and for the accountability of public funds.

In regard to the cited papers we have now studied the issue of mortality in respect of AAA that is the focus of these papers and we contend that they are not of material substance in the context of our paper and that it would therefore be inappropriate to draw references to them. We set out below our reasoning.

The Potential Impact of Abdominal Aortic Aneurysm (AAA) on aggregate hospital mortality.

It is worth stating again that our paper is the first to analyze overall hospital mortality across countries over a long period of time, and establishes differences not only in levels but, more importantly, in trends. As Reviewer 2 has already confirmed this is an important and novel contribution to knowledge.

Reviewer 1 is in effect claiming that a single cause of death (AAA) is important to understanding the aggregate data and trends. In support of that the Reviewer has previously drawn attention to differences in the timing of treatment policies in the two countries we study (Scotland and England) in relation to this particular cause of death.

There are of course many causes of death that might contribute to what we have observed and it is for future research to disentangle which of these might account for the differential trends. However, we can easily establish that AAA as a cause of death cannot account for what we have observed to any substantial degree. This is simply a matter of numerical magnitudes.

The data we report concern all deaths in hospital for Scotland and England separately. These total approximately 240,000 a year in England (the range is actually 260,000 in 2005/06 to 224,000 in 2012/13). The published paper that the Reviewer 1 cites (<http://www.nejm.org/doi/full/10.1056/NEJMoa1600931>) indicates that deaths that could be attributed to AAA can be no more than 5,000 a year in the period 2005-2012. Hence AAA can at most constitute 2% of deaths in hospital.

The point here is that a condition that is undoubtedly potentially and actually fatal – and thus results in a very high risk of death in hospital – is nevertheless comparatively rare and so cannot account for the observed differences in aggregate mortality that we report.

If we were to draw attention to this particular cause of death it would simply be to state that it cannot be an explanation of the challenge we have identified. However, taking that approach would simply risk distracting readers from the overall picture that we wish to describe.

None of the studies that are referred to by the Reviewer come close to our approach either in terms of scope (health system vs single cause) or method (time series analysis of individual patient records accounting for confounders). They are thus not germane to our paper.

Of course we accept that different perspective on an issue are valuable and may result in differences of opinion as to what it is relevant to include or exclude from any one scientific paper. However, we have stated our reasons for omitting this strand of previous research from our paper and think it must be an editorial decision as to which view should prevail.

We would like to end by pointing out that one Reviewer was entirely supportive of our paper, and that we believe this work has the potential to generate great interest and debate in a topic that has not been previously addressed and which raises important questions regarding hospital care and treatments in the United Kingdom.

Thank you in anticipation

VERSION 3 – REVIEW

REVIEWER	Janet Powell Imperial College London No Competing Interest
REVIEW RETURNED	19-Jul-2017

GENERAL COMMENTS	I am very disappointed that the authors are unwilling to even mention potential clinical reasons for differences between Scotland and England in the discussion. To avoid any possible misunderstanding, I will be very plain. The discussion needs to acknowledge that there are known potential clinical confounders eg timing of screening (big impact on AAA-related mortality, see MASS trial results or Wanhainen et al Circulation 2016) and use of EVAR (big impact on elective mortality). It is only the discussion which requires revision. In addition to my previous comments, I would add that it is possible that in-hospital mortality, for aneurysm ruptures, may not even have exactly the same meaning in England and Scotland depending on whether moribund patients are formally admitted to hospital (not always the case in England). This underscores the importance of acknowledging that variations in clinical practice, especially the known variations, might contribute to the findings. Given the choice of submission journal, this paper is meant to be read by clinicians, so it seems especially important to acknowledge the potential clinical confounders/contributors.
---

VERSION 3 – AUTHOR RESPONSE

Reviewer 1

Comment

I am very disappointed that the authors are unwilling to even mention potential clinical reasons for differences between Scotland and England in the discussion. To avoid any possible misunderstanding, I will be very plain. The discussion needs to acknowledge that there are known potential clinical confounders eg timing of screening (big impact on AAA-related mortality, see MASS trial results or Wanhainen et al Circulation 2016) and use of EVAR (big impact on elective mortality). It is only the discussion which requires revision.

Response

We regret the reviewer’s disappointment and have previously set out our reasons for not going into details regarding clinical differences in a paper that seeks to establish the evidence regarding aggregate trends that remains to be explained. As requested we have now extended the discussion to make explicit reference to these differences in a paragraph which explains that the trends in in-hospital mortality observed might have different clinical causes, including, for example, different timing of introduction of screening programmes for AAA and have made explicit reference to the suggested paper.

Comment

In addition to my previous comments, I would add that it is possible that in-hospital mortality, for aneurysm ruptures, may not even have exactly the same meaning in England and Scotland depending on whether moribund patients are formally admitted to hospital (not always the case in England). This underscores the importance of acknowledging that variations in clinical practice, especially the known variations, might contribute to the findings. Given the choice of submission journal, this paper is meant to be read by clinicians, so it seems especially important to acknowledge the potential clinical confounders/contributors.

Response

As noted above we have now acknowledged such clinical factors in the additional paragraph in the discussion.

VERSION 4 – REVIEW

REVIEWER	Janet Powell Imperial College London No Competing Interest
REVIEW RETURNED	02-Sep-2017

GENERAL COMMENTS	Better, since on page 10 you now acknowledge at least one clinical factor, ie differential timing of screening programmes. It would be really helpful if you could add the obvious second clinical confounder (you do use the term confounders in the plural) by introducing the phrase "and the differential use of endovascular repair". Why? Endovascular repair extends the range of patients who can be offered elective surgery, to minimise the numbers of older patients refused elective repair, and reduce operative mortality 3-fold in comparison with open repair. It is well known that Scottish vascular surgeons were slower than those in England to adopt endovascular repair.
---

VERSION 4 – AUTHOR RESPONSE

Reviewer 1

Comment

Better, since on page 10 you now acknowledge at least one clinical factor, ie differential timing of screening programmes.

It would be really helpful if you could add the obvious second clinical confounder (you do use the term confounders in the plural) by introducing the phrase "and the differential use of endovascular repair".

Why? Endovascular repair extends the range of patients who can be offered elective surgery, to minimise the numbers of older patients refused elective repair, and reduce operative mortality 3-fold in comparison with open repair. It is well known that Scottish vascular surgeons were slower than those in England to adopt endovascular repair.

Response

We have changed the reference regarding the introduction of AAA screening programmes to two separate references, one for each country, which include the dates when the screening programmes were rolled out in each country; we also added a sentence regarding the use of endovascular repair.